# The Effects of Plant Growth-Promoting Bacteria with Biostimulant Features on the Growth of a Local Onion Cultivar and a Commercial Zucchini Variety

Giorgia Novello [1], Patrizia Cesaro [1,*], Elisa Bona [2], Nadia Massa [1], Fabio Gosetti [3], Alessio Scarafoni [4], Valeria Todeschini [2], Graziella Berta [1], Guido Lingua [1] and Elisa Gamalero [1]

1 Dipartimento di Scienze e Innovazione Tecnologica, Università del Piemonte Orientale, Viale T. Michel, 11-15121 Alessandria, Italy; giorgia.novello@uniupo.it (G.N.); nadia.massa@uniupo.it (N.M.); graziella.berta@uniupo.it (G.B.); guido.lingua@uniupo.it (G.L.); elisa.gamalero@uniupo.it (E.G.)
2 Dipartimento di Scienze e Innovazione Tecnologica, Università del Piemonte Orientale, Piazza Sant'Eusebio, 5-13100 Vercelli, Italy; elisa.bona@uniupo.it (E.B.); valeria.todeschini@uniupo.it (V.T.)
3 Dipartimento di Scienze dell'Ambiente e della Terra, Università degli Studi di Milano-Bicocca, Piazza della Scienza, 1-20126 Milano, Italy; fabio.gosetti@unimib.it
4 Dipartimento di Scienze per gli Alimenti, la Nutrizione e l'Ambiente, Università degli Studi di Milano, Via Celoria, 2-20133 Milano, Italy; alessio.scarafoni@unimi.it
* Correspondence: patrizia.cesaro@uniupo.it; Tel.: +39-(0)131360244

**Abstract:** The reduction of chemical inputs due to fertilizer and pesticide applications is a target shared both by farmers and consumers in order to minimize the side effects for human and environmental health. Among the possible strategies, the use of biostimulants has become increasingly important as demonstrated by the fast growth of their global market and by the increased rate of registration of new products. In this work, we assessed the effects of five bacterial strains (*Pseudomonas fluorescens* Pf4, *P. putida* S1Pf1, *P. protegens* Pf7, *P. migulae* 8R6, and *Pseudomonas* sp. 5Vm1K), which were chosen according to their previously reported plant growth promotion traits and their positive effects on fruit/seed nutrient contents, on a local onion cultivar and on zucchini. The possible variations induced by the inoculation with the bacterial strains on the onion nutritional components were also evaluated. Inoculation resulted in significant growth stimulation and improvement of the mineral concentration of the onion bulb, induced particularly by 5Vm1K and S1Pf1, and in different effects on the flowering of the zucchini plants according to the bacterial strain. The present study provides new information regarding the activity of the five plant growth-promoting bacteria (PGPB) strains on onion and zucchini, two plant species rarely considered by the scientific literature despite their economic relevance.

**Keywords:** plant growth-promoting bacteria; *Pseudomonas* sp., flowering time; nutritional profile; *Allium cepa*; *Cucurbita pepo*

## 1. Introduction

The latest report on the global State of *Food Security and Nutrition* was released in 2020 by the United Nations. According to this document: "Almost 690 million people suffered from hunger in 2019, that is 10 million more than in 2018 and slightly less 60 million more in five years". The percentage of undernourished people is about 9% of the world population. A constant upward trend of this parameter has been observed in the last five years, clearly indicating that the food demand related to food availability has increased proportionally with the growth of the world population (http://www.fao.org/3/ca9692en/online/ca9692en.html#; Last access: 04/29/2021). Consequently, several approaches have been proposed to solve or alleviate this issue, such as a fair distribution of food resources, avoiding unnecessary food waste, which, in Europe,



reaches 88 Million tonnes (Mt) per year [1], or supporting the consumption of zero km products [2].

In such a context, environmental sustainability has become an essential requirement. To minimize the side effects for human and environmental health, reducing the use of chemical inputs (i.e., fertilizers and pesticides) is mandatory. This is a target widely shared by both farmers and consumers. For instance, the use of nitrogen and phosphorus based fertilizers boosts the eutrophication process, and the excessive and unregulated usage of irrigation water causes a dramatic reduction of aquifer and river flows, particularly in both arid and semiarid climatic regions where intensive farming is practiced [3,4]. In addition, consumers are increasingly aware of such issues and they demand healthy, nutritious foods produced according to sustainable environmental practices.

Among the various possible strategies, the use of plant biostimulants (PBs) has become increasingly important and this is demonstrated by the fast growth of their global market (2241 million dollars in 2018) and by the increased rate of registration of new PBs (+12.5% from 2013 to 2018) [5].

Although different definitions of plant biostimulants have been given, the most recent provided by the EU Regulation 2019/1009 was, "*A plant biostimulant shall be an EU fertilising product the function of which is to stimulate plant nutrition processes independently of the product's nutrient content with the sole aim of improving one or more of the following characteristics of the plant or the plant rhizosphere: i) nutrient use efficiency, ii) tolerance to abiotic stress, iii) quality traits, or iv) availability of confined nutrients in the soil or rhizosphere*" [6]. Therefore, PBs, including a variety of substances (protein hydrolysates, humic and fulvic substances, and animal and vegetal protein extracts) as well as plant beneficial microorganisms, including plant growth-promoting bacteria (PGPB), satisfy the requirements stated in the previous definition [7]. Mechanisms at the base of the PGPB effects are classified as direct and indirect. Direct mechanisms include the provision of nutrients, such as nitrogen (nitrogen fixation by free-living or symbiotic bacteria), phosphorus (via phosphate solubilisation), and iron (through the production of low molecular weight chelating molecules called siderophores) and the synthesis of phytohormones (auxins, cytokinins and gibberellins). Indirect mechanisms involve plant disease suppression mediated by the synthesis of siderophores, antibiotic and lytic enzymes, the stimulation of the plant immune system, and the enhancement of plant tolerance to environmental stresses. All these bacterial strategies have proven to be effective against plant diseases caused by viruses, bacteria, fungi, and plant-parasitic nematodes [8–12].

Plant inoculation with PGPBs meets the two main expectations of modern agriculture: the increase of plant yield, especially in field conditions where PGPBs often show inconsistent performance, and the improvement of the nutritional value of seeds and fruits [13]. Several papers demonstrated a positive influence of PGPBs on the quality of fruits in different plant species, such as maize, tomatoes, peaches, strawberries, and beans [14–20].

In this work, five bacterial strains (*Pseudomonas fluorescens* Pf4, *P. putida* S1Pf1, *P. protegens* Pf7, *P. migulae* 8R6, and *Pseudomonas sp.* 5Vm1K) were selected from our collection according to their plant growth promotion capability and their effects on fruit/seed nutrient contents. The first aim of this work was to characterize the effects of these five PGPBs with suitable features as biostimulants on the growth of a local cultivar of onion (*Allium cepa* L.) and on zucchini (*Cucurbita pepo* L.). Onion has great economic importance: in 2017, its production in Europe and Italy, respectively, was 10,429,425 and 410,536 tons. At the European level, Spain is the main zucchini producer. In 2020 in Italy, 15,937 ha of cultivated land produced 411,300.5 tons of zucchini (ISTAT, 2020). We focused our attention on a well-known and common commercial variety of zucchini (Altea, Syngenta) and on a typical onion cultivar cultivated in Castelnuovo Scrivia (Alessandria, Italy) characterized by a red-purple bulb and a sweet taste, which was recently labeled as PAT ("Prodotto Agroalimentare Tradizionale" which means "Traditional Agrifood Product"),as an Italian food quality recognition. Taking into account the typicality of this local

product, the second goal was the assessment of the possible variations induced by the inoculation with the bacterial strains on the onion nutritional component (both the mineral and vitamin contents).

## 2. Materials and Methods

### 2.1. Origins of Bacterial Strains and Seeds

*Pseudomonas fluorescens* Pf4, *P. putida* S1Pf1, *P. protegens* Pf7, *P. migulae* 8R6, and *Pseudomonas sp.* 5Vm1K were employed for plant inoculation. In our previous studies, we demonstrated that *P. fluorescens* Pf4 and *Pseudomonas sp.* 5Vm1K increased the yield and modulated the soluble sugars, organic acids, and vitamin contents in strawberry plants cultivated under reduced fertilization [15]. The strain Pf4, co-inoculated with arbuscular mycorrhizal fungi (AMF), promoted maize growth, enhanced the yield in field conditions, and increased the grain starch content, particularly the digestible fraction [14]. *P. protegens* Pf7 is a PGPB on reed (*Phragmites australis*) (data not shown) and positively cooperates with AMF increasing the corn size and the number of flowers as well as the anticipated flowering time in saffron (*Crocus sativus* L.) [21]. This bacterial strain improved the denitrification efficiency in the treatment of wastewater [22]. *P. putida* S1Pf1 is a PGPB of tomatoes (data not shown) and is able to increase the tolerance of *Chysanthemum carinatum* to CY phytoplasma [23]. Thanks to its oxygenase activity, the strain S1Pf1 has been used in the degradation of 1,5-naphthalenedisulfonic acid in aqueous solutions coupled to UV-photolysis [24]. Finally, *P. migulae* 8R6 was kindly provided by Bernard Glick. This is a bacterial endophyte that is able to promote the growth of tomatoes in both stressed and non-stressed conditions [25] and to increase the tolerance of perwincle to Flavescence dorée phytoplasma [26]. Some of the bacterial physiological traits involved in plant growth promotion were previously assessed, while others were characterized in this work according to the methods reported in the next paragraph.

Seeds of *Allium cepa* L. were provided by Consorzio di Tutela della cipolla rossa e dorata di Castelnuovo Scrivia, while seeds of *Cucurbita pepo* L. (cv. Altea) were bought by Syngenta.

### 2.2. Assessment of PGPB Physiological Traits

The 1-aminocyclopropane-1-carboxylate (ACC) deaminase activity was measured according to the method described by Penrose and Glick [27] with a standard curve of $\alpha$-ketobutyrate between 0.05 and 0.5 $\mu$mol.

The siderophore production was evaluated on Chrome Azurol S (CAS) agar according to the method of Schwyn & Neilands [28]. The bacterial strains were inoculated at the center of each plate and incubated at 28 °C for seven days. The ability to produce siderophores was indicated by the occurrence of a yellow-orange halo around the colony and was measured with a caliper as the ratio between the two diameters of the halo and the two diameters of the colony.

Phosphate solubilization was evaluated using two different media: one containing dicalcium phosphate (DCP) and one containing tricalcium phosphate (TCP). The strains were inoculated in the center of each plate and incubated at 28 °C for 15 days. DCP solubilization was indicated by a clarification halo around the colony; TCP solubilization was identified by colony growth on the medium [29].

A qualitative assessment of the ability to synthesize indole-3-acetic acid (IAA) was performed following the method of de Brito et al. [30]. The bacterial strains were inoculated at the centre of a trypticase soy agar (TSA) plate with 10% L-tryptophan (5 mM) added. A nitrocellulose disc was then placed on the agar medium and incubated at 28 °C for 3 days. The membrane was then dipped in Salkowsky's reagent (FeCl₃ 2% in perchloric acid 35%), and a red/pink halo around the colony indicated a positive reaction.

The IAA production was quantified according to the method of Forni et al. [31]. The bacterial strain was inoculated in 20 mL of M9 salt minimal medium added to sucrose at

10 mM and L-tryptophan (400 mg mL$^{-1}$; Fluka) and incubated on a rotary shaker at 150 rpm in the dark at 28 °C for 4 days. The bacterial suspension was centrifuged at 4500 rpm and 4 °C for 20 min. Two milliliters of Salkowsky's reagent was added to 1 mL of the supernatant. After 30 min of incubation, the amount of IAA produced was evaluated at $\lambda$ 530 nm using an IAA solution (100 mg mL$^{-1}$) as the standard.

All these tests were conducted in triplicate.

### 2.3. Onion and Zucchini Cultivation

Onion and zucchini seeds were surface sterilized with 10% sodium hypochlorite for 5 min and washed five times with sterilized water for 7 min. The seeds were pregerminated on moist sterile filter paper at 24 °C for 5 days. One germinated seed was sown in each pot. Pre-germinated seeds were inoculated with a 10$^8$ CFU/mL suspension of each bacterium and transferred in 3 L sterile pots, one plantlet per pot containing 33% of quartz sand and 66% of sterilized soil (Acid peat, pH 6.0, electric conductivity 0.30 dS m$^{-1}$, dry apparent density 100 Kg m$^{-3}$, and total porosity 85% *v/v*). One set of plants was not inoculated and was used as a control. Each treatment (Control, Pf4, Pf7, S1Pf1, 8R6 and 5Vm1K) comprised ten pots as independent replicates. The plants were watered three times a week and fed with organic fertilizer (Algaren Twin, Green Has Italia, Canale d'Alba (CN), Italy).

From the beginning of flowering, the zucchini plants were monitored every two days; the number of female and aborted fruits as well as the fruit number and size were recorded.

After four months, the onion plants were harvested. The roots were separated from the shoots, the leaves were cut, and the fresh weight of bulbs was recorded. Bulbs of *A. cepa* were measured (major and minor diameter), cut into cubes, frozen in liquid nitrogen and stored at −80 °C for further analyses.

### 2.4. Mineral, Vitamin and Sulphur Compound Analysis

Ten grams of the onion bulb samples was accurately weighed and extracted with 5.0 mL of a mixture of acetone/ethanol/water 50/40/10 (*v/v/v*). The extraction was favoured by using a vortex for 5 min, and then the mixture was centrifuged at 7000 rpm for 5 min. The liquid fraction was filtered on a 0.2 μm polytetrafluoroethylene (PTFE) filter (Phenomenex, Milan) and directly subjected to UHPLC-MS/MS analysis.

The LC/MS analyses were performed using a Nexera Liquid Chromatography Shimadzu (Kyoto, Japan) system equipped with a DGU-20A3R Degasser, two LC-30AD Pumps, a SIL-30AC Autosampler, a CTO-20AC column compartment and a CMB-20A Lite system controller. The system was interfaced with a 3200 QTrapTM LC–MS/MS system (AB Sciex S.r.l., Milano, Italy) by a Turbo VTM interface equipped with an ESI probe. The 3200 QTrapTM data were processed by Analyst 1.5.2 software (Toronto, ON, Canada).

The standard stock solutions (each at concentration of 100.00 mg L$^{-1}$) were prepared in methanol and preserved in dark conditions (−20 °C).

The chromatographic column was a Kinetex C18 (2.1 mm × 100 mm, 2.6 μm, Phenomenex, Italy). The mobile phase was a mixture of ammonium acetate (1.0 mM) in water (A) and MetOH (B) both with the addition of 0.1% formic acid, eluting at flow rate 0.400 mL min$^{-1}$ in the following gradient conditions: 0.0–1.0 min from 30 to 50% B, 1.0–1.1 min 100% B, 5.0 min 100% B, 5.1–8.0 min 30% B. The injection volume was 10.0 μL and the oven temperature was set at 40 °C.

Turbo ion spray (TIS) ionization was performed using the Turbo VTM interface working in positive ion mode. The parameters were set as follows: curtain gas (N$_2$) at 30 psig; nebulizer gas GS1 and GS2 at 40 and 45 psig, respectively; desolvation temperature (TEM) at 500 °C; collision activated dissociation gas (CAD) at 6 units of the arbitrary scale of the instrument; and ion spray voltage (IS) at +5000 V.

A unit mass resolution was established and maintained in each mass-resolving quadrupole by keeping a full width at half maximum (FWHM) of about 0.7 u.

All the analytes presented many transitions: for each species the most intense transition was used for the quantitative analysis and referred to as the "quantifier" transition, while the second one (the "qualifier" transition) was employed in the identification step, as a confirmation. The "quantifier" and "qualifier" transitions and the optimal instrumental potential values for each compound are reported in Table 1.

**Table 1.** The "quantifier" and "qualifier" transitions and the optimal instrumental potential values for Alliin, vitamin C, and Retinol.

| Analyte | Q1 | Q3 | Dwell Time (ms) | DP (V) | EP (V) | CEP (V) | CE (V) | CXP (V) |
|---------|-----|-------|-----------------|--------|--------|---------|-----------|---------|
| Alliin | 178 | 88/74 | 25 | 14.80 | 4.0 | 15.1 | 12.1/28.4 | 2.2/2.2 |
| Vitamin C | 177 | 95/141 | 25 | 23.00 | 4.3 | 15.1 | 16.0/9.3 | 2.2/2.3 |
| Retinol | 269 | 93/91 | 25 | 38.86 | 3.5 | 17.7 | 31.4/63.0 | 2.1/2.1 |

## 2.5. Water Content

About one gram of each sample was finely chopped with a razor blade, placed in a small glass capsule, weighed with an analytical balance (Mettler-Toledo ME54) and then heated to 110 °C until a constant weight was reached. Before each measurement, capsules were closed with a lid and let to cool to room temperature in a sealed jar containing silica gel as desiccant. The samples were analyzed in triplicate.

## 2.6. Sugar Concentration

Sucrose, D-glucose, and D-fructose were spectrophotometrically investigated (Lambda II, PerkinElmer, Milan, Italy) using an enzymatic kit (R-Biopharm, Darmstadt, Germany, cat. 10716260035).

## 2.7. Mineral Concentration

Samples (0.25 g) were ground with dry ice and digested using a microwave digestor system (Multiwave-Eco, Anton Paar GmbH, Graz, Austria), in Teflon™ tubes with 10 mL of 65% $HNO_3$. A two-step power ramp was applied (step 1:200 W in 10 min, maintained for 5 min; step 2:650 W in 10 min, maintained for 15 min). The samples were diluted 1:40 with Milli-Q water and the ion concentration was measured by inductively coupled plasma mass spectrometry (ICP-MS) (Bruker AURORA M90 ICP-MS).

## 2.8. Protein Extraction and Quantification

Samples (1 g) were ground in a mortar with dry ice. Four mL of extraction solution (10% trichloroacetic acid (TCA), 50% acetone, 1% polyvinylpyrrolidone, and 2% β-mercaptoethanol) was added and incubated for 30 min at room temperature under vigorous shaking. The suspensions were kept overnight at −20 °C to allow protein precipitation. Following centrifugation at 12,000 *g*, the precipitates were rinsed twice with 5 mL of cold acetone and allowed to dry at room temperature in a fume hood. The pellet was then solubilized in a solution containing urea (7 M), thiourea (2 M) and 2% CHAPS.

The concentration of the proteins in solution was determined according to the Bradford assay [32]. BSA dissolved in the solubilizing buffer was used as the standard.

## 2.9. Protein Electrophoresis

Sodium Dodecyl Sulphate—PolyAcrylamide Gel Electrophoresis (SDS-PAGE) was performed according to Laemmli [33] on 12% polyacrylamide gels using a MiniProtean III device (BioRad). The electrophoretic separations were carried out at constant current

(16 mA for each gel). After the runs, the polypeptides were visualized by silver staining according to Shevchenko et al. [34].

### 2.10. Statistical Analysis

The data normality and homogeneity of the variance were checked using the Shapiro–Wilk and Levene tests, respectively. Based on the obtained results, parametric (ANOVA) or non-parametric (Kruskal–Wallis or ARTANOVA) one-way tests were applied to compare different groups (Control, Pf4, Pf7, S1Pf1, 8R6 and 5Vm1K), followed by post hoc tests (the LSD Fisher's test and multiple comparisons of means with *p*-values adjusted according to Bonferroni and Dunn test or permutation test, respectively) with the cut-off at $p < 0.05$. Statistical analyses were performed using R (v. 3.5.1) [35].

## 3. Results

### 3.1. Plant Beneficial Physiological Traits of PGPB Strains

The characterization of the physiological activities of the strains Pf4, S1Pf1, 8R6 and 5Vm1K was mostly described in the papers cited in Table 2. The strain Pf7, previously identified as *P. fluorescens*, has been now recognized through the analysis of the whole genome (unpublished results) as *P. protegens*. This strain is able to produce IAA and siderophores and solubilize organic phosphate at acid, neutral and alkaline pH (data not shown) but not in the inorganic forms (Table 2).

**Table 2.** Physiological trait characterization of the bacterial strains employed to inoculate onion and zucchini seeds.

| Strain | ACCd [1] | Siderophore [2] | Phosphate Solubilization (DCP) [3] | Phosphate Solubilization (TCP) [4] | IAA [5] | References |
|---|---|---|---|---|---|---|
| *P. fluorescens* Pf4 | 0.0 ± 0.0 | 3.8 ± 0.2 | 1.25 | + * | 103 ± 2 | Berta et al., 2014 [14] |
| *P. protegens* Pf7 | ND | 1.8 ± 0.6 * | 0.00 | + * | 39.0 ± 0.7 * | This work |
| *P. putida* S1Pf1 | 0.0 ± 0.0 | 5.5 ± 0.2 | 0.00 | + * | 10.6 ± 0.6 | Gamalero et al., 2010 [36] |
| *P. migulae* 8R6 | 10.90 | 1.9 ± 0.3 * | 0.00 | + * | 39.43 | Rashid et al., 2012 [37] |
| *Pseudomonas* sp. 5Vm1K | ND | 3.8 ± 0.2 | 0.9 | + * | ++++ [6] | Bona et al., 2015 [15] |

[1] ACC deaminase evaluate as alpha-ketobutyrate µmol mg$^{-1}$ h$^{-1}$; [2] Siderophore production was assessed on universal Chrome Azurol S (CAS) agar [28] after 7 days of culture and measured as the ratio between the diameter of the halo (HD) and that of the colony (CD); [3] Phosphate solubilization was assessed according to Goldstein [29], on dicalcium phosphate (DCP) and measured as the ratio between the diameter of the halo (HD) and that of the colony (CD); [4] Phosphate solubilization was assessed according to Goldstein [29], on tricalcium phosphate (TCP): the bacterial growth indicates a positive reaction; [5] Indole-3-acetic acid (IAA) production was quantified according to Forni et al. [31] and measured as µg IAA ml$^{-1}$ h$^{-1}$; [6] IAA assessment has been done with a qualitative test according to de Brito et al. [30]; * data obtained in this work; ND = not determined.

### 3.2. Effects of PGPBs on the Onion and Zucchini Growth and Yield

The effect of the PGPBs on onion growth was evaluated on the onion bulb biomass and size. The bulb fresh weight was increased by each bacterial strain: *P. fluorescens* Pf4 (+80.4%), *P. protegens* Pf7 (+94.0%), *P. putida* S1Pf1 (+140.6%), *P. migulae* 8R6 (+132.4%) and *Pseudomonas* sp. 5Vm1K (+200.9%) (Figure 1A). Consistently, both the minor and major diameters of the bulbs were higher in PGPB-inoculated plants compared with in the uninoculated controls (Table 3). The strains S1Pf1 and 5Vm1K were the highest performing plant growth promoters for onion. The effect of the PGPBs on onion bulbs is shown in Figure 1B.

**Table 3.** Morphometric parameters of the bulb of *Allium cepa* inoculated or not (Control) with different bacterial strains (*Pseudomonas fluorescens* Pf4, *P. protegens* Pf7, *P. putida* S1Pf1, *P. migulae* 8R6, *Pseudomonas sp.* 5Vm1K). Mean value ± standard error. Different letters in the row indicate significant differences among the treatments according to ANOVA followed by LSD Fisher's test ($p < 0.05$).

| | **Control** | **Pf4** | **Pf7** | **S1Pf1** | **8R6** | **5Vm1K** |
|---|---|---|---|---|---|---|
| Collar diameter [A] (cm) | 1.0 ± 0.2 a | 1.3 ± 0.2 ab | 1.0 ± 0.2 a | 1.5 ± 0.1 b | 1.4 ± 0.1 b | 1.6 ± 0.2 b |
| Bulb major diameter [B] (cm) | 3.5 ± 0.1 a | 4.4 ± 0.4 b | 4.7 ± 0.3 bc | 5.2 ± 0.2 c | 5.2 ± 0.2 c | 5.3 ± 0.2 c |
| Bulb minor diameter [C] (cm) | 3.3 ± 0.2 a | 4.1 ± 0.4 b | 4.4 ± 0.3 b | 5.0 ± 0.2 bc | 4.8 ± 0.2 bc | 5.5 ± 0.2 c |
| Water content (%) | 90.4 | 82.6 | 90.9 | 90.4 | 90.3 | 90.8 |

[A] The collar diameter was measured at the base of the onion shoot, at the level at which it is inserted in the bulb; [B, C] The major and minor diameters were the two diameters of the ellipsoid, the geometric figure to which the shape of an onion can be generalized.

The yield of zucchini plants was evaluated as the number of fruits, number of aborted fruits, fruit length and number of female flowers. Although the fruit weight and size were unaffected by the PGPB inoculation (Figure 2A,B), the number of fruits was increased only by the strain *Pseudomonas* sp. 5Vm1K (10 fruits in 5Vm1K inoculated plants vs. two fruits in uninoculated control plants, which is +400%) at the third sampling time (Figure 1B and Figure 3).

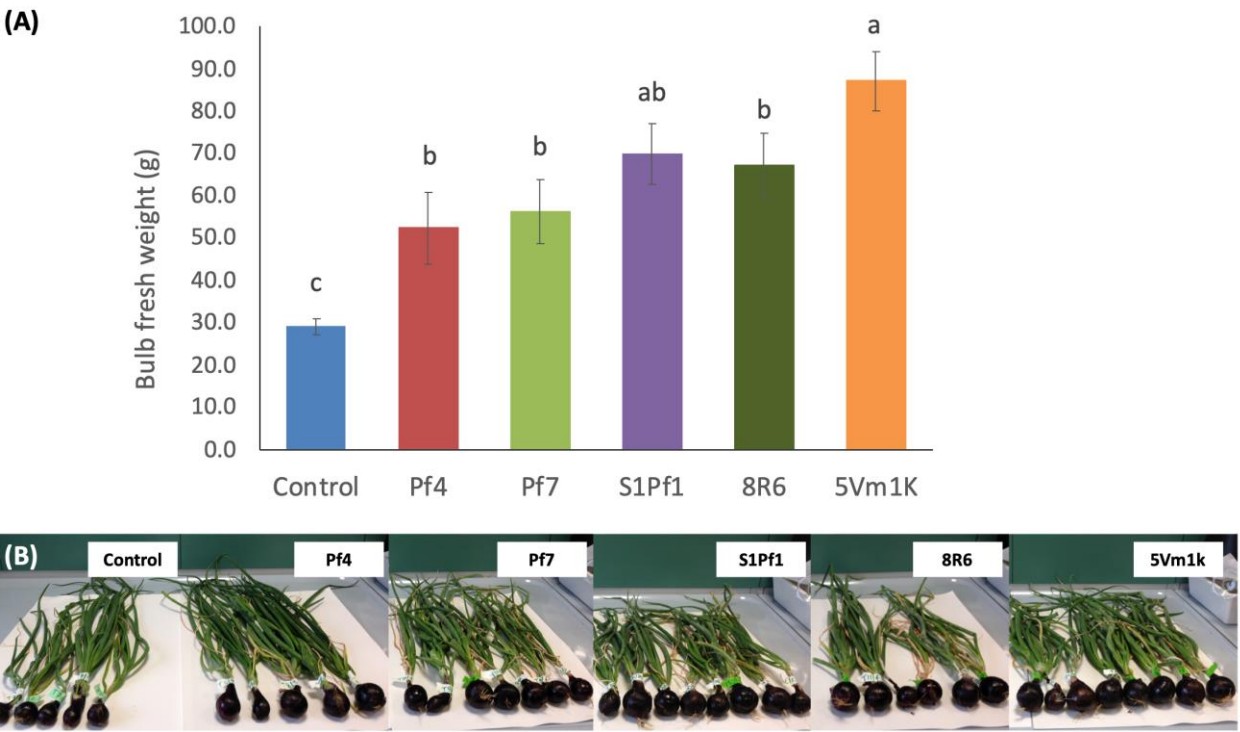

**Figure 1.** Effects of the five PGPB strains on onion bulb weight (**A**) and growth (**B**). Different letters on the bars in the histogram indicate significant differences among the treatments according to ANOVA followed by the LSD Fisher's test ($p < 0.05$).

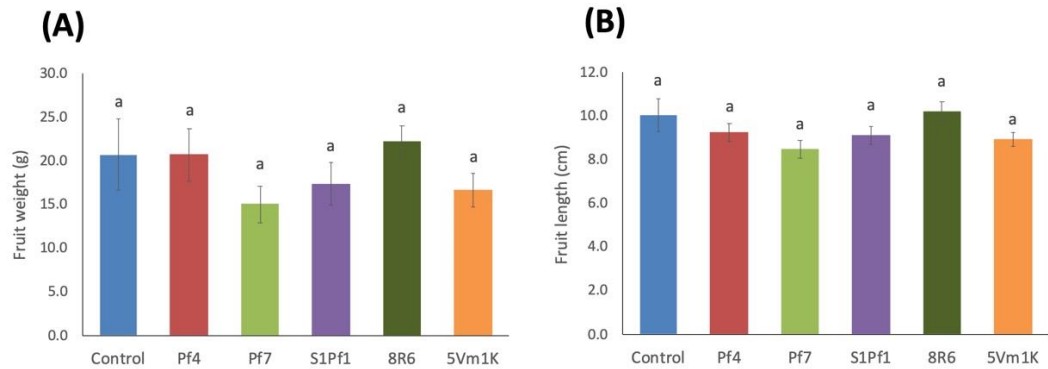

**Figure 2.** Effects of the five PGPB strains on the zucchini fruit weight (**A**) and length (**B**). Different letters upon the bars in the histograms indicate significant differences among the treatments according to the Kruskal–Wallis test followed by the Dunn test ($p < 0.05$).

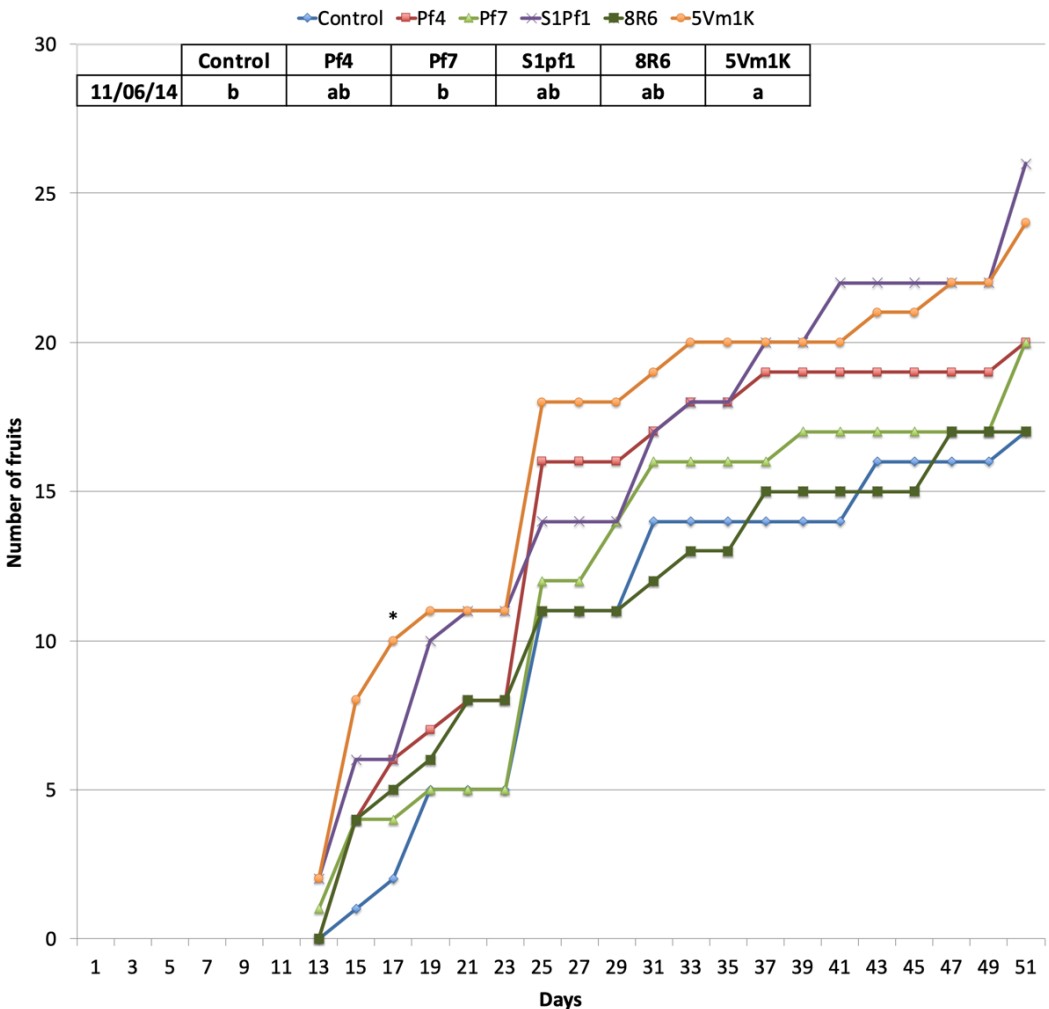

**Figure 3.** The number of fruits produced by zucchini plants inoculated or not with the five PGPBs during two months of growth. The * indicates a sampling time where significant differences among the treatments were observed; in the table, the different letters indicate significant differences among the treatments according to the Kruskal–Wallis test followed by the Dunn test ($p < 0.05$).

No significant differences were recorded for the number of aborted fruits (Figure 4). However, the number of female flowers monitored every two days revealed that at the second and third sampling time the number of flowers increased in plants inoculated with

the strains *P. putida* S1Pf1 (+145.5% and +34%, respectively) and *Pseudomonas* sp. 5Vm1K (+200% and 56.1%, respectively) compared to controls. Starting from the fourth sampling time the trend of the curves representing the production of female flowers were very similar for all the plant treatments. However, at the 18th and 24th sampling times, a decrease in the flower production was observed in plants inoculated with the strains Pf4, S1Pf1, 8R6 and 5Vm1k compared to uninoculated controls (Figure 5).

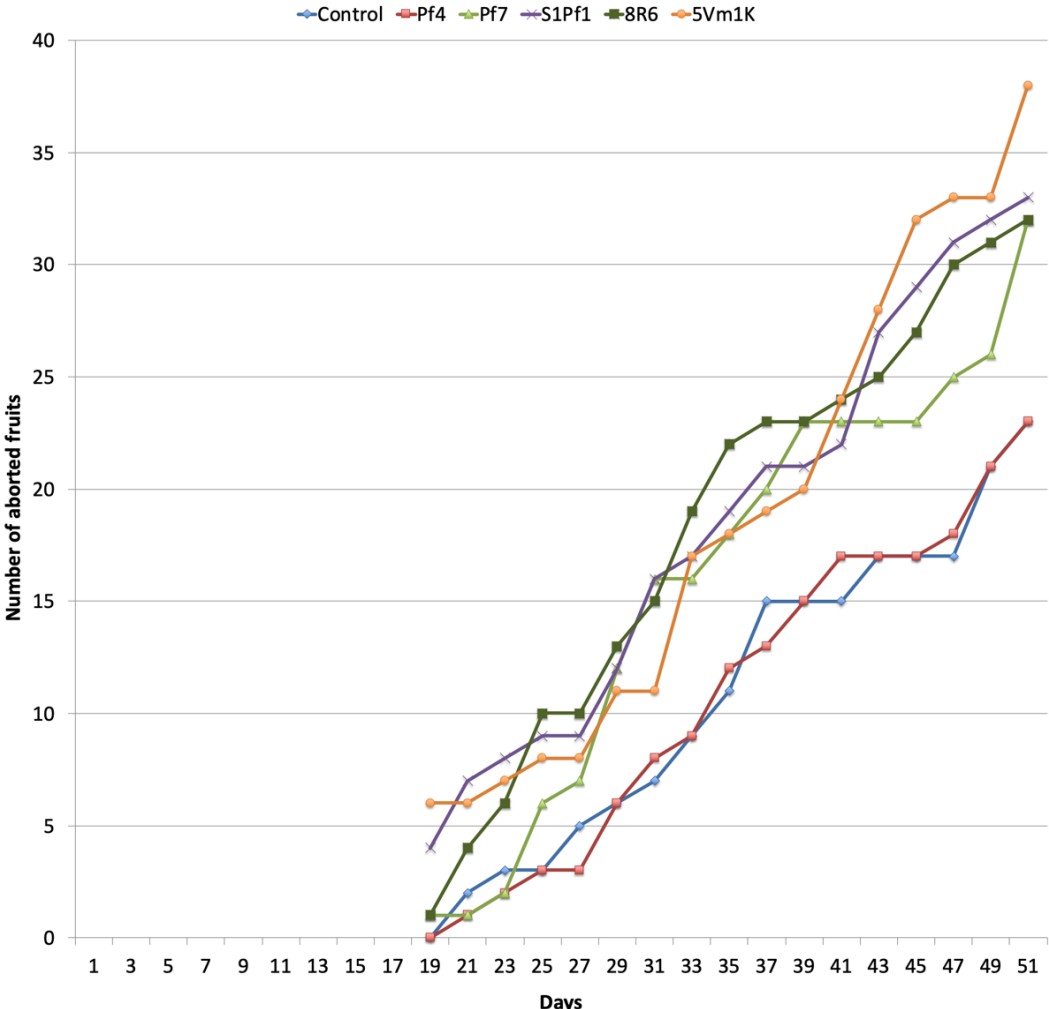

**Figure 4.** The number of aborted fruits produced by zucchini plants inoculated or not with the five PGPBs during two months of growth. The * indicates a sampling time where significant differences among the treatments were observed; in the table the different letters indicate significant differences among the treatments according to ANOVA followed by multiple comparisons of the means with the *p*-values adjusted according to the Bonferroni or to Kruskal–Wallis test followed by the Dunn test ($p < 0.05$).

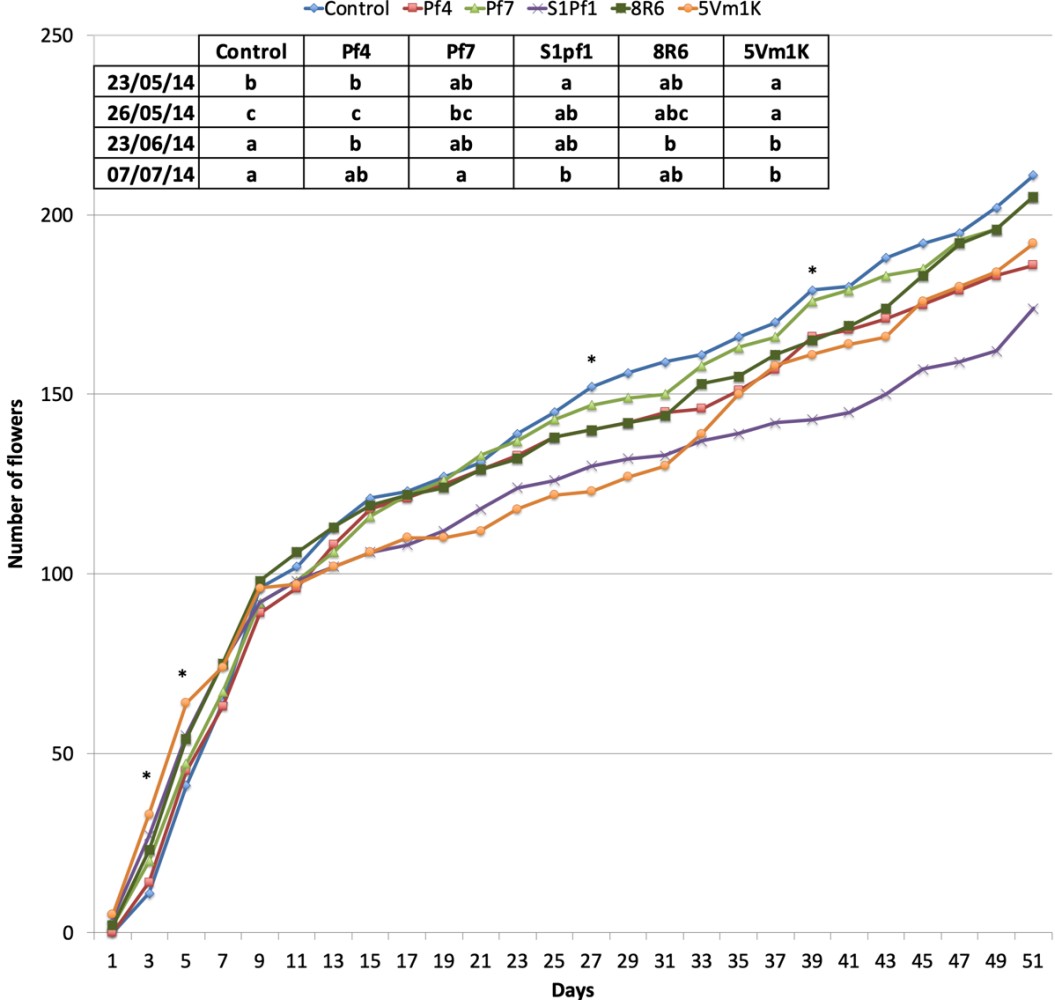

**Figure 5.** The number of female flowers produced by zucchini plants inoculated or not with the five PGPBs during two months of growth. The * indicates a sampling time where significant differences among the treatments were observed; in the table, the different letters indicate significant differences among the treatments according to ARTANOVA followed by a permutation test or to the Kruskal–Wallis test followed by the Dunn test ($p < 0.05$).

### 3.3. Effects of PGPBs on the Onion Quality

The nutritional components of the onions were evaluated as the protein content and sucrose, glucose, fructose, alliin, vitamin C, retinol and mineral concentration. No statistically significant differences ($p > 0.05$) were found in the onion bulbs regarding either the protein content or the sugar concentration (sucrose, glucose and fructose) (Table 4). Both alliin and vitamin C remained unaffected by the plant inoculation with PGPBs. The retinol concentration was below the detection limit of the method for all the plant treatments (Table 4). Interestingly, the five PGPBs modulated the mineral contents of the onion bulb. With the exception of Ca, which was reduced following the inoculation with 5Vm1K, the other macronutrients were unaffected by inoculation. However, the PGPB inoculation induced significant variations of in the micronutrient concentration (Table 5).

The levels of copper, zinc and cadmium in onion bulbs were reduced by all the bacterial strains (Table 5). Similarly, the amount of Mn was reduced by all the PGPB strains except S1Pf1. On the contrary, the selenium concentration was increased in onion bulbs treated with PGPBs. The strains 8R6 and 5Vm1K showed the most intense effects on the micronutrients. In detail, copper was reduced by 44.6% and 41.4%, zinc by 60.4% and 56.3%, and cadmium by 71.1% and 74% after inoculation with *P. migulae* 8R6 and *Pseudomonas* sp. 5Vm1K, respectively.

On the contrary, seed bacterization with 8R6 and 5Vm1K led to an increased selenium concentration in onion bulb by +20.0% and +31.33%, respectively (Table 5).

**Table 4.** The alliin, vitamin C, and retinol concentrations of the bulb of *Allium cepa* inoculated or not (Control) with different bacterial strains (*P. fluorescens* Pf4, *P. protegens* Pf7, *P. putida* S1Pf1, *P. migulae* 8R6, and *Pseudomonas* sp. 5Vm1K). n.d. = not determined, the value was under the detection level for the method. Mean value ± standard error. Different letters in the row indicate significant differences among the treatments according to ANOVA followed by the LSD Fisher's test ($p <$ 0.05).

| | **Control** | **Pf4** | **Pf7** | **S1Pf1** | **8R6** | **5Vm1K** |
|---|---|---|---|---|---|---|
| Proteins | 1.1 ± 0.1 a | 0.9 ± 0.5 a | 1.1 ± 0.1 a | 1.5 ± 0.1 a | 1.1 ± 0.1 a | 1.2 ± 0.1 a |
| Sucrose | 5.6 ± 0.4 a | 6.0 ± 0.2 a | 5.8 ± 0.3 a | 6.3 ± 0.5 a | 6.2 ± 0.3 a | 5.9± 0.1 a |
| D-glucose | 14.3 ± 0.7 a | 12.9 ± 0.6 a | 14.8 ± 0.5 a | 13.2 ± 1.2 a | 14.4 ± 0.6 a | 12.7 ± 0.9 a |
| D-fructose | 12.5 ± 0.9 a | 12.7 ± 0.7 a | 13.9 ± 0.4 a | 14.4 ± 0.6 a | 12.6 ± 0.5 a | 13.9 ± 0.7 a |
| Alliin ($\mu$g g$^{-1}$) | 1.22 ± 0.16 a | 10.13 ± 9.26 a | 9.13 ± 7.60 a | 7.07 ± 2.63 a | 0.80 ± 0.08 a | 1.07 ± 0.09 a |
| Vitamin C ($\mu$g g$^{-1}$) | 3.2 ± 0.8 a | 2.3 ± 0.2 a | 2.8 ± 0.2 a | 2.9 ± 0.2 a | 2.1 ± 0.2 a | 2.9 ± 0.3 a |
| Retinol ($\mu$g g$^{-1}$) | n.d. | n.d. | n.d. | n.d. | n.d. | n.d. |

**Table 5.** Mineral concentrations in the bulb of *Allium cepa* inoculated or not (Control) with different bacterial strains (*P. fluorescens* Pf4, *P. protegens* Pf7, *P. putida* S1Pf1, *P. migulae* 8R6, and *Pseudomonas sp.* 5Vm1K). The mean and standard errors are presented ($n$ = 3). Different letters in the row indicate significant differences among the treatments according to ANOVA followed by the LSD Fisher's test ($p <$ 0.05).

| | **Control** | **Pf4** | **Pf7** | **S1Pf1** | **8R6** | **5Vm1K** |
|---|---|---|---|---|---|---|
| Na (mg g$^{-1}$) | 1.89 ± 0.06 a | 1.90 ± 0.24 a | 2.09 ± 0.06 a | 1.66 ± 0.05 a | 1.96 ± 0.08 a | 2.14 ± 0.13 a |
| Mg (mg g$^{-1}$) | 9.04 ± 0.50 a | 9.70 ± 0.50 a | 8.41 ± 0.71 a | 10.19 ± 0.45 a | 8.87 ± 0.40 a | 9.02 ± 1.16 a |
| K (mg g$^{-1}$) | 166 ± 7 a | 157 ± 6 a | 160 ± 6 a | 157 ± 5 a | 100 ± 42 a | 159 ± 6 a |
| Ca (mg g$^{-1}$) | 18 ± 3 ab | 19 ± 3 ab | 12 ± 1 bc | 25 ± 3 a | 12 ± 4 bc | 8 ± 1 c |
| Mn ($\mu$g g$^{-1}$) | 108 ± 7 a | 82 ± 5 b | 80 ± 4 b | 114 ± 8 a | 83 ± 1 b | 85 ± 12 b |
| Fe ($\mu$g g$^{-1}$) | 114 ± 2 a | 120 ± 12 a | 118 ± 15 a | 111 ± 4 a | 97 ± 5 a | 125 ± 21 a |
| Cu ($\mu$g g$^{-1}$) | 39.12 ± 1.95 a | 29.51 ± 1.89 b | 26.22 ± 1.87 bc | 25.43 ± 1.35 bc | 21.67 ± 2.45 c | 22.94 ± 0.42 c |
| Zn ($\mu$g g$^{-1}$) | 261 ± 7 a | 180 ± 15 b | 131 ± 14 cd | 162 ± 13 bc | 103 ± 5 d | 114 ± 11 d |
| Se ($\mu$g g$^{-1}$) | 1.15 ± 0.11 c | 1.36 ± 0.04 ab | 1.22 ± 0.06 b | 1.36 ± 0.08 ab | 1.38 ± 0.05 ab | 1.51 ± 0.03 a |
| Cd ($\mu$g g$^{-1}$) | 2.04 ± 0.21 a | 1.00 ± 0.09 b | 0.79 ± 0.06 bc | 1.1 ± 0.11 b | 0.59 ± 0.02 c | 0.53 ± 0.10 c |

The electrophoretic patterns in the SDS-PAGE analysis were similar in all samples (Figure 6).

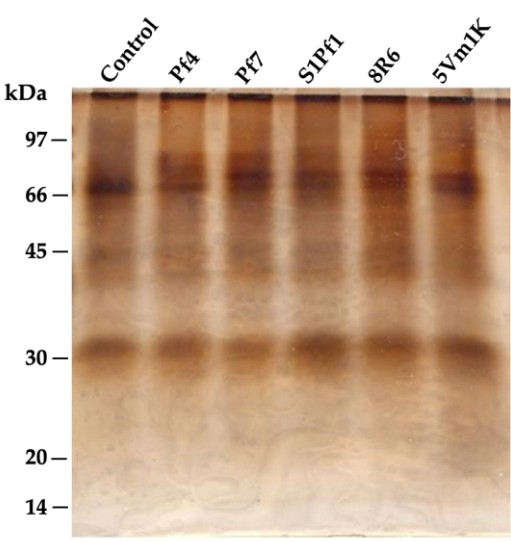

**Figure 6.** SDS-PAGE analysis of total proteins extracted from onions inoculated or not (Control) with different bacterial strains (Pf4, Pf7, S1Pf1, 8R6, 5Vm1K).

## 4. Discussion

The bacterial strains used in this work have been previously well characterized for their plant growth-promoting capabilities and/or involvement in soil/water assisted phytoremediation. The enhancement of plant development was often combined to an important modulation of the nutritional components of fruits (maize, strawberries and tomatoes) [13–16,18,38]. Due to their proven beneficial effect on plants, all the considered PGPBs, except *P. migulae* 8R6, have been sold to companies producing biofertilizers. When approaching the procedure to market a bacterial strain as a biostimulant or biofertilizer, there are different possible issues. Among them, plant host specificity involves a large number of bacterial and plants properties: the capability to colonize and to possibly promote the growth of a certain group of plants is based on recognition mechanisms, chemotactic properties, metabolic abilities, and rhizospheric competence. On the other hand, tight control of the beneficial bacterial traits regarding plant compounds (exuded molecules), the properties of the root surface, and sensitivity to bacterial phytohormones all determine the success of the plant–bacteria interaction [39]. Based on these ideas, in this paper we aimed to assess the effect of these five well established PGPBs on two other plant species, one belonging to a widely marketed cultivar (zucchini) and the other to a local niche cultivar (onion). Despite the great diffusion of this plant species, the impact of PGPBs on zucchini and onion has not been explored yet. In fact, using "plant growth-promoting rhizobacteria" and "zucchini/cucurbita pepo/summer squash/courgette" as keywords we did not find any paper on the main scientific databases and using "onion" and "plant growth-promoting rhizobacteria" we obtained only 14 papers with different degrees of relevance.

The PGPB capabilities of the bacterial strains that we previously observed were confirmed in onions, where the bulb size and weight were increased by the bacterial strains, especially by *Pseudomonas* sp. 5Vm1K and *P. putida* S1Pf1. This is consistent with the results obtained using the free-living nitrogen fixers *Azospirillum* and *Azotobacter* inoculated in onion cultivated with the recommended amount of nitrogen (N,) phosphorus (P) and potassium (K) fertilizers. Researchers reported that the inoculation of *Azotobacter* along with 100% NPK improves vegetative growth, while *Azospirillum* along with 100% NPK improved yield and nutritional parameters of onions, for instance, the ascorbic acid and total sugar contents [40]. Similarly, strains of *Azotobacter* sp., *Sphingobacterium* sp., and *Burkholderia* sp. increased both the growth parameters and yield attributes of onions, with the triple inoculation treatment being the most efficient [41].

Looking at the onion nutritional components 1 kg contains about 88 g of carbohydrates, 12 g of proteins, 1.8 g of fats, 274 mg of calcium, 367 mg of phosphorous, 4.2 mg iron, 0.3 mg thiamine (vitamin B1), 0.2 mg riboflavin (vitamin B2), 1.2 mg niacin (vitamin B3) and 76 mg of ascorbic acid (Vitamin C) [42]. The typical onion pungency is related to the volatile oil allyal propyl disulphide [43].

Alliin is the precursor of allicin, which is a bioactive molecule that is released when fresh onion or garlic is cut or crushed. Allicin has possible applications in both medicine and agriculture. In addition to having anti-inflammatory, anticancer, and antithrombotic properties, this molecule modulates the immune system and is active against Gram-negative and Gram-positive bacteria as well as against multidrug-resistant pathogens, such as methicillin-resistant *Staphylococcus aureus* (MRSA) [44]. On the other hand, seedlings of *Arabidopsis thaliana* exposed to allicin were inhibited in the primary root elongation in a concentration-dependent manner with a MIC of 75 μM [45]. Taking into consideration these properties, we measured the content of alliin in onion bulbs produced by the different plant treatments. Unfortunately, although in certain treatments with PGPBs (Pf4, Pf7 and S1Pf1) an increasing trend of the content of alliin was recorded, the data were highly heterogeneous and did not allow to statistically significant differences to be highlighted.

Similarly, the protein, sugar and vitamin C concentration in onion bulbs in inoculated and uninoculated plants did not change. However, interesting results regarding the mineral composition of the onion bulb were obtained. While the macronutrients (Mg, K, and Ca) were not affected by PGPB inoculation, the level of the micronutrients was strongly modulated by the bacterial strains, with the exception of iron which instead showed no significant variations. The amount of copper and zinc was reduced by all the PGPBs, while that of selenium was increased. All these latter four are considered essential trace elements that support an effective immune response [46]. Selenium is a cofactor of enzyme involved in the thyroid hormone metabolism and is integral part of many antioxidant enzymes [47]. Notably, cadmium dramatically decreased upon bacterial inoculation. Cadmium is a toxic heavy metal that may contribute to several diseases in human and animals and can be absorbed by plants destined for the food supply. Its presence in soil and water is caused by industrial and agricultural activities [48]. The reduction of cadmium in our samples could be due by the microorganism activity inducing the production of chelator polypeptides such as phytochelatins [49,50].

The protein profile of onion bulbs evaluated by SDS-PAGE analysis was similar for all plant treatments.

The overall data obtained clearly indicate an improvement in the onion yield parameters (bulb weight, size, and diameter) without significant variation of the concentration of the organics molecules that can contribute to an organoleptic character. However, it should be considered that this onion cultivar is a local production with an official Italian food quality recognition; in this context, PGBPs that increase the yield while maintaining the typical organoleptic characteristics can be viewed positively. This is an important issue in Europe, and particularly in Italy, were the food types and authenticity are protected by specific laws. The European Union established three levels of recognition for food, including Protected Designation of Origin (PDO), Protected Geographical Indication (PGI), and Traditional Specialties Guaranteed (TSG). In addition, the Optimal Quality Terms (OQT) "mountain product" and "product of island farming" were introduced (1151/2012 EU Regulation) in order "to support the promotion and protection of the regional foods with special quality, to avoid food frauds and to become aware of, and adopt good agricultural practices" [51,52].

The effects of PGPBs on zucchini plants was less evident than those observed in onions. The main result obtained was an anticipation of the flowering time and the increase of the number of flowers by two strains, *P. putida* S1Pf1 and *Pseudomonas* spp. 5Vm1K compared to the controls, at the first stages of flowering. This is consistent with the earlier flowering time observed in strawberry plants cultivated under reduced fertilization conditions and inoculated with the strain 5Vm1K together with a mixed mycorrhizal inoculum [15]. Similar effects were reported for strawberries after inoculation with the KI-2 strain KI-2 of *Bacillus cereus* [53]. The process of flowering results from balanced carbon and nutrient requests by the different plant organs [54]. According to Bona et al. [15], the flowering process follows two steps: the accumulation of the right amount of nutrients and phytohormones in the shoot apex and the transformation to an inflorescence apex. Among the phytohormones involved, gibberellins, which are tetracyclic diterpenoids consisting of isoprene residues, are well known to have a role in seed germination, elongation of the shoot and flowering and fruit setting [55]. The synthesis of gibberellic acid by PGPBs belonging to the genera *Achromobacter*, *Azospirillum*, *Agrobacterium*, *Azotobacter*, *Bacillus*, *Herbaspirillum*, *Clostridium*, *Burkholderia*, *Gluconobacter*, *Pseudomonas* and *Rhizobia* has been reported [56]. An earlier flowering indicates that the shift from the allocation of resources for plant growth to the allocation of these nutrients for reproduction comes sooner. A shift to earlier flowering leads to faster plant development, and therefore the increase of the flowering rate is also a desired trait for ornamental and officinal plants [57,58]. However, the anticipation of flowering can also have costs, especially when plants do not invest as many resources on other functions, such as growth and defence, leaving the plants more susceptible to herbivory or disease or less efficient in producing flowers or seeds [59]. In

fact, in the second half of the experimental time, the total number of flowers produced by zucchini plants inoculated with the strains *P. putida* S1Pf1 and *Pseudomonas* spp. 5Vm1K were significantly lower than the number produced by the uninoculated control plants. Similarly, the number of aborted fruits was higher in plants inoculated with these two bacterial strains, although not significantly.

Overall, this work allowed us to obtain new information regarding the activity of these five PGPB strains on two plant species, onion and zucchini, which are rarely considered by the scientific literature in spite of their economic relevance. By considering the previous and current data on these five bacterial strains, the issue of PGPB host specificity is evident. However, this is only one of the factors determining the bottlenecks in PGPB commercialisation [60]. Other parameters, such as the safety of the bacterial strain for human health, the survival capability of the bacterial cells in open field conditions, the possible improvement of their beneficial capabilities when combined in mixed inoculant, the choice of a commercial formulation that is practical for farmers and, at the same time, allows the highest rate of survival, and the mode and time of the biofertilizer distribution, need to be seriously considered before commercialization. However, the advantages given by the application of PGPBs over chemical fertilizers and/or pesticides are evident when considering the increase of the plant yield and biocon-trol of soil-borne plant diseases, which can largely overcome the cost required by the pro-cedures for the selection, identification, characterization and formulation of new biofertilizers.

**Author Contributions:** Conceptualization, E.B. and E.G.; Methodology, E.B., A.S., and F.G.; Software, N.M.; Investigation, G.N., E.B., A.S., P.C., and V.T.; Resources, G.B. and G.L.; Data Curation, N.M.; Writing—Original Draft Preparation, E.G., G.N., N.M. and P.C.; Writing—Review & Editing, E.G., G.L. and G.B.; Funding Acquisition, G.B. All authors have read and agreed to the published version of the manuscript.

**Funding:** This research received no external funding.

**Institutional Review Board Statement:** Not applicable.

**Informed Consent Statement:** Not applicable.

**Acknowledgments:** The authors want to thank Bernard Glick who provided the strain *P. migulae* 8R6.

**Conflicts of Interest:** The authors declare no conflict of interest.

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
