# Peer review of "The Effects of Plant Growth-Promoting Bacteria with Biostimulant Features on the Growth of a Local Onion Cultivar and a Commercial Zucchini Variety"

_agronomy, doi:10.3390/agronomy11050888_

Round 1
Reviewer 1 Report
This manuscript assessed the effects of plant growth promoting bacteria (PGPB) (Pf4, Pf7, S1Pf1, 8R6, 5Vm1K) on onion and zucchini. The results indicated that bacterial strains could stimulate onion bulb growth, especially S1Pf1 and 5Vm1K, and changed mineral concentration of onion. Furthermore, the flowering time of the zucchini inoculated with PGPBs were different from controls. The methodological strategy of this manuscript is adequate, and might provide candidate strains to enhance crop production. However, the Title and Introduction could be improved. And then, the important results should be added in Abstract. Some other comments that should be addressed are listed below:
- Line 29: ‘PGPB’ should be revised to be ‘plant growth promote bacteria (PGPB)’
- Line 48-49: Rephrase this sentence, which was highly similar with Abstract (Line 18-19).
- Line 108: Rephrase ‘modulated’ to ‘modulate’.
- Line 130: ‘ACC’ should be ‘Aminocyclopropane (ACC)’
- Line 132-136: The method of quantification of siderophore is inaccurate by measurement of the halo and colony delimiters. It would be better to use the methods of reference ‘Siderophore Production by Bacillus megaterium: Effect of Growth Phase and Cultural Conditions’ (DOI: 10.1007/s12010-013-0562-y).
- Line 143: ‘IAA’ should be ‘indole-3-acetic acid (IAA)’.
- Line 149: ‘Forni et al. [31]’ should be ‘Forni et al. [31].’.
- Line 182: ‘PTFE’ should be ‘polytetrafluoroethylene (PTFE)’.
- Line 266: The title of Table 2 should be ‘Plant growth promoting characterization of the bacterial strains employed to inoculate onion and zucchini plant’.
- Line 272-273: The strains being positive capability of tricalcium phosphate solubilization could present transparent zone in medium added Ca3(PO4)2. Please carefully check the bacterial growth whether represents a positive reaction.
- Line 274-275: The order of references [31] and [30] should be adjusted.
- The descriptions of plant beneficial physiological traits of strains Pf4, S1Pf1, 8R6 and 5Vm1K should be added in 3.1 part.
- Figure 1: The figure 1B-F should be combined into one picture and should better to use one control picture.
- The methods of significance analysis of figure and table should be explained.
- Table 3: Rephrase ‘maior’ to ‘major’; add the standard error of water content; explain the difference of three diameter (Collar diameter, Bulb major diameter, and Bulb minor diameter).
- Line 322: Rephrase ‘Pseudomonas spp.’ to ‘Pseudomonas sp.’
- Line 323-325: Rephrase this sentence.
- Table 4: ‘n.d.’ should be explained.
- Line 356: The manganese of onion inoculated with S1Pf6 was increased in comparison with control.
- Line 366: Rephrase ‘raw’ to ‘row’.
- Combine Line 368 with Line 337-346.
- The x-axis of figure 2-5 should be described as the number of days instead of specific date, and the layout of figures need to be adjusted.
- Line 381: Add ‘,’ in front of ‘it’s possible’.
- Line 391-394, 409-410, 438-439: Rephrase these sentences academically.
Author Response
We send the responses and the manuscript with track changes

Reviewer 2 Report
This a straight forward study of the effects of several PGPB on the growth of onion and zucchini. The effects are well documented. My only reservation is that these are pot studies using peat moss to grow the plants. How applicable are these results to plants grown in the field.
Author Response

(The authors gave the same response as above.)

Reviewer 3 Report
This article focuses his studies on improving the yield of onion and zucchini crops in pot trials. The variables studied are considered complete and well explained and executed. Here are some considerations.
Background considerations:
1. As in this type of microbial ecology tests, and in order to be able to evaluate the real capacity of the growth-promoting bacteria to exert their effects on the growth and development of the plant, an "artificial" substrate is chosen. It is well explained in the text that there are more variables that can affect the results obtained, but it would be good to include if it is valued to transfer this same test to field conditions, in experimental plots, for example. Also if the realization of bacterial consortia is valued.
2. The authors talk about PGPB. Inoculation is carried out at the pre-germinated seed level and colonization is established at the root level. Why was the term PGPBacteria used instead of PGPRhizobacteria? In my opinion, it would be more appropriate and accurate to talk about the second.
3. In the conclusions it is commented that the strains, which come from previous duly cited tests, must be safe for human and environmental health. Seeing that tests are still being carried out in different plant species, wouldn't it be advisable to sequence them to find out their genetic information and detect the presence of virulence genes? This is not something that should be included in this article, as it is a complete essay on the sidelines. However, perhaps it would be good to value it and leave it reflected in this text. Especially in case promising results continue to be obtained that can be overshadowed by the presence of virulence genes that complicate their release into the environment.
4. In section 2.3. the composition of the substrate used in the test "33% of quartz sand and 66% of sterilized soil (acid peat)" is discussed. Some authors consider that the use of sterilized soil can produce certain volatile compounds. This could affect bacteria and growth. How was the peat sterilized? It is recommended to provide bibliographic citations in this regard.
Format considerations:
1. Line 239: move the title to the next page.
2. Line 240: replace "one gram" with "1 g", to preserve the text format.
3. Line 251: define what the acronym "SDS-PAGE" means.
4. Figure 1. It is recommended to enlarge the size of the graph A).
5. Figure 2. It is recommended to adjust the size of the graphs so that they both fit in a single row (parallel to the horizontal). It is recommended to include the letters of the "analysis of variance" above the error bars to know the significant differences between treatments.
6. Figure 4. I would assess whether to leave it. It does not provide information as long as no change has been observed in the number of aborted fruits. In the legend remove, if the graph is left, the meaning of "*", since this is not the case.
7. Line 346: At the end, refer to "Table 5" of micronutrients.
The authors have done a great job, very comprehensive and interesting.
Author Response

(The authors gave the same response as above.)
